# Effects of Different Parts of the Rose Flower on the Development, Fecundity, and Life Parameters of *Frankliniella occidentalis* (Pergande) (Thysanoptera: Thripidae)

**DOI:** 10.3390/insects14010088

**Published:** 2023-01-13

**Authors:** Ding-Yin Li, Dan Zhou, Jun-Rui Zhi, Wen-Bo Yue, Shun-Xin Li

**Affiliations:** 1Institute of Entomology, Guizhou Provincial Key Laboratory for Agricultural Pest Management of the Mountainous Region, Guizhou University, Guiyang 550025, China; 2Qingzhen City Agricultural and Rural Bureau, Guiyang 551400, China

**Keywords:** *Frankliniella occidentalis*, rose petals, kidney bean pods, rose flowers, life table

## Abstract

**Simple Summary:**

The western flower thrips *Frankliniella occidentalis* (Thysanoptera: Thripidae) is a major pest of vegetables and flower crops worldwide, and different parts of rose flowers can provide different nutrients to thrips, which may have different nutritional effects on *F. occidentalis*. In this study, experiments were performed to investigate the influence of rose petals, rose flowers, and 10% honey solution + kidney bean pods on the individual life-history traits and on the population growth of *F. occidentalis*.

**Abstract:**

*Frankliniella occidentalis* (Pergande) is an important horticultural pest that causes serious damage to rose plants, which is one of its preferred foods. In this study, rose petals, rose flowers, and 10% honey solution + kidney bean pods were chosen as foods to assess their influence on the growth, development and fecundity of *F. occidentalis*. The results showed that developmental time of immature *F. occidentalis* with the following trend: rose flowers <10% honey solution + kidney bean pods < rose petals < kidney bean pods. The longevities of both female and male adults were lowest when feeding on the rose petals and were highest when feeding on rose flowers. The fecundity was in the following order: rose flowers >10% honey solution + kidney bean pods > rose petals > kidney bean pods. The net reproductive rate (*R*_0_), intrinsic rate of increase (*r*), and finite rate of increase (*λ*) of *F. occidentalis* feeding on rose petals and kidney bean pods were lower than those feeding on rose flowers and 10% honey solution + kidney bean pods. The development, longevity, fecundity, and parameters have significantly changed since F_1_ generation after feeding with the three food types. The results indicated that different parts of rose flowers had a significant effect on the development of thrips, and nectar and pollen had a positive effect on thrips population increase and reproduction.

## 1. Introduction

Different food types have different nutrient components, which have significant effects on the morphology, growth, development, reproduction, and population dynamics of insects [1,2,3]. da Silva [4] found that centesimal composition of components in the diets was correlated significantly with larval and pupal stages and/or pupal weight of *Helicoverpa armigera*. Wen et al. [5] found that the larval development duration of *Trabala vishnou gigantina* was negatively correlated with the protein content in the host plant, and that the oviposition of female adults was negatively correlated with the soluble sugar content in the host plant. Gao et al. [6] reported that three *Adelphocoris* spp. had significantly higher nymphal developmental and survival rates, along with increased adult longevity and fecundity after feeding on flowering cotton and alfalfa than on non-flowering plants of either species. Angelella and Riley [7] revealed that the addition of pine pollen shortened the growth and development time of *Frankliniella fusca* and increased its oviposition. Fu et al. [8] found that the addition of honey solution and tea pollen was beneficial to *Thrips hawaiiensis* growth, development, and reproduction.

*Frankliniella occidentalis* (Pergande), western flower thrips (WFT), is a globally distributed horticultural crop pest. It originated in North America and spread rapidly to other countries around the world [9,10]. WFT was first found on pepper in the Haidian District of Beijing in June 2003; subsequently, it spread to Yunnan, Zhejiang, Guizhou, and other provinces in China [11,12]. In recent years, WFT has developed as a major pest of flowers and vegetables and caused great economic losses in China [12,13]. WFT feeds on flowers and the young parts of the host plant, such as tender leaves and buds, resulting in spots on the feeding sites and shrinkage of growth points [10]. WFT can also spread several viruses, which significantly harms crop production [14,15,16]. Thrips prefer flowers to leaves, as flowers provide more nutrients and promote their development and fecundity [17,18]. A previous study reported that the WFT population increased almost exponentially during the plant flowering period [19]. Several studies have demonstrated that adding pollen to plant leaves had a significant effect on WFT growth, development, and reproduction [20,21].

WFT, as a flower-inhabiting insect, feeds predominantly on flowers and prefers *Rosa rugosa* flowers [22]. Cao et al. [23,24] also reported that WFT had a higher population on *R. chinensis* flowers than that on *Gardenia jasminoides* flowers. Previous studies have shown that WFT caused the most damage when roses were in bloom, as breakouts occurred during this period [25,26]. Shen et al. [27] found that pollen of rose, camellia, rape, and 10% sucrose solution could promote oviposition in WFT. Recently, Qi et al. [28] found that rose species could influence the population and physiological performance of WFT, and that digestive enzymes might play an important role in nutrient metabolism and growth of thrips. Earlier studies have tested the flower promotional effect on WFT. However, knowledge regarding the influence of different rose flower parts on WFT development, reproduction, and life parameters is scarce. We hypothesized that different rose flower parts will have different effects on WFT and that nectar and pollen will play an important role in triggering WFT population. To reveal their influence, in this study, rose petals and rose flowers (containing petal, pollen, and nectar) were selected as the experimental diets, and kidney bean (*Phaseolus vulgaris*) pods were used as the control. Simultaneously, to evaluate the effect of honey (i.e., a supplemental food), a 10% honey solution + kidney bean pods was also selected as test material. The findings of this study will provide useful evidence regarding the relationship between dietary rose flower and WFT population, as well as provide a basis to understand the rose flower triggers of WFT population outbreaks.

## 2. Materials and Methods

### 2.1. Insect Source and Foods

WFT (*F. occidentalis*) were collected from kidney bean plants in Huaxi District, Guiyang City, Guizhou Province, China, and raised with kidney bean pods in an artificial climate box (RXZ series multi-stage programmable intelligent artificial climate box Ningbo Jiangnan Instrument Factory, Ningbo, China) for more than 50 generations. The rearing conditions were as follows: temperature was kept at 25 ± 1 °C, relative humidity (RH) of 70 ± 5%, and photoperiod (light:dark) of 14 h:10 h.

The roses, *R. rugosa* of the dark red variety, were purchased from the Yunnan Xinhaihui flower industry base (Kunming, China). The rose petals and flowers were used as food. In the group-rearing experiment, the different kinds of foods were set as follows. For the rose petal diet, only the petals were retained, and the other flower parts were removed. For the rose flower diet, stamens and pistils containing pollen and nectar were kept along with 5 flower petals. For the supplementary diet (honey solution), rape flower honey was purchased from supermarkets (produced by Jieshi Co. Ltd., Shanghai, China), and 100 mL honey was added to 900 mL distilled water to obtain a 10% honey solution (*v*/*v*). The kidney bean pods were purchased from a farmer’s market and soaked in water with detergent for 10 min, then washed with running water several times and air-dried for use. For the 10% honey solution + kidney bean pods, the rinsed and dried kidney bean pods were soaked in 10% honey solution for 10 min and air-dried again for use.

In the individual thrips development experiment, the following different food types were used as the test materials: rose petal (just one rose petal), rose flower (one flower petal with all stamens and pistils), 10% honey solution + kidney bean pod (dried kidney bean pod soaked in honey solution was cut into 1.5 cm section), and kidney bean pod (1.5 cm section of kidney bean pod).

### 2.2. The Group-Rearing of WFT on Different Foods

Approximately 200 WFT comprising 3-day enclosed male and female adults reared on kidney bean pods for more than 50 generations were placed in a box (20 cm × 13 cm × 9 cm in width, breadth, and height, respectively) with three different food types (rose flowers, rose petals, 10% honey solution + kidney bean pods) and allowed to lay eggs. The adults were removed after 12 h oviposition, and the different food materials with eggs (eggs laid in the plant tissue) were transferred into another clean insect box. After the eggs hatched, the larvae were recorded as the first instar larvae of the F_1_ generation. These WFT larvae were continuously reared to adults, which were recorded as F_1_ adults. Some of F_1_ adults were collected by a suction implement for the individual development and reproductive experiment; the remaining F_1_ adults were also collected but inoculated into a new box with the same type of food from which they previously emerged and adults were allowed to lay eggs. After 12 h oviposition of F_1_ adults, the different foods with eggs were removed and placed in another clean insect box to continuously rear to F_2_ adults. The same steps were repeated until the F_3_ generation adults. The new foods were replaced every 2 days during the group rearing. The F_1_, F_2_, and F_3_ adults were used as test materials for the following experiment. Thrips were reared by the same type of food in all 3 generations. Adults of WFT reared on kidney bean pods were used as control groups.

### 2.3. Development Duration of WFT

Approximately 200 thrips comprising 3-day enclosed male and female adults in F_1_, F_2_, and F_3_ generations after feeding with the three different food types, and adults reared on kidney bean pods for more than 50 generations were placed in a box (20 cm × 13 cm × 9 cm in width, breadth, and height, respectively) with four different food types (rose flowers, rose petals, 10% honey solution + kidney bean pods, and kidney bean pods), and thrips were allowed to lay eggs. After 12 h oviposition, the foods containing eggs were removed, placed in another clean insect box, and observed twice a day (08:00 and 20:00). After the eggs hatched, the newly hatched first instar larvae were individually placed in a transparent insect raising cup (bottom diameter 2.9 cm; top diameter 3.7 cm; and height 3.3 cm) with the same food types from which they reared and absorbent filter paper at the bottom of the cup. The development time and survival of each instar were observed under the microscope and recorded twice a day (08:00 and 20:00), and fresh food was replaced every other day until thrips emerged into adults. A total of 60 thrips for each treatment were recorded in every generation, and the treatments were repeated in three replicates. Thrips were kept on the same diet in the same treatment group, and they were reared in an artificial climate box, in which the conditions were maintained at a temperature of 25 ± 1 °C, RH of 70 ± 5%, and a 14:10 h L:D photoperiod.

### 2.4. The Fecundity of WFT

Newly emerged thrips (one female and one male of F_1_, F_2_, and F_3_ adults from Section 2.3) were paired and placed in a transparent insect cup with the same type of food from which they were reared as specified in Section 2.3; the cups were placed in an artificial climate box (the conditions were the same above). The life spans of the adults were recorded daily until the adult died. The food containing thrips eggs was transferred daily to a new transparent insect cup and reared for over 6 d to ensure that all eggs hatched. The number of hatched larvae represented the number of eggs laid [29], as the fecundity of thrips. The hatched larvae were reared on the same food types from which they fed until the adult sex determination. These reproduction assays were conducted with three replicates of 20 pairs of thrips per treatment. Based on the development, survival of each life-stage, fecundity of WFT feeding on different foods, and the principle of the age-stage, two-sex life tables were constructed using the method of Chi and Liu [30] and Chi [31].

### 2.5. Data Analysis

Microsoft Excel 2016 software was used to analyze the original data. Two-sex MSChart software [32] (available for free download at http://140.120.197.173/Ecology/ accessed on 3 February 2021.) was used to calculate the age-stage specific survival rate (*s_xj_*), age-stage life expectancy (*e_xj_*), age-stage reproduction value (*v_xj_*), age-specific survival rate (*l_x_*), and life table parameters of different generations of WFT fed on different foods, where *x* represents the age divided by survival time, and *j* represents the development stage [33,34]. Average values and standard errors of each life stage, including the pre-adult stage (egg to adult emergence), development times, life parameters at different generations, and feeding on different foods, were determined using a 100,000-time bootstrapping method. The significance of differences in developmental duration, fecundity, and life parameters of WFT was analyzed using a paired bootstrap test at the 5% significance level (two-sex MSChart) [35].

## 3. Results

### 3.1. Effects of Different Food Types on the Development of WFT in Successive Generations

When WFT fed on the same food, the development durations of WFT were distinctive among control and three consecutive generations (Table 1). Regardless of the food types, the durations of the egg, first instar, second instar, prepupa, and pupa stages were significantly shorter than those in WFT fed on the control (kidney bean pods) since the F_1_ generation, and there were no significant differences among F_1_, F_2_, and F_3_ generations except for that of the first instar thrips fed on rose petals and 10% honey solution + kidney bean pods. The immature periods of the WFT fed on the three food types were also shorter than those of the control, although the immature period decreased in successive generations; however, the degree of decrease was not different among different food types.

For WFT feeding on different foods in the same generation, the development of each stage was also different (Table 1). Egg-stage durations of WFT fed on different food types increased in the following order: rose flowers <10% honey solution + kidney bean pods < rose petals in all three generations. The durations of the first instar larval stage were similar to the durations of the egg stage in the F_1_ and F_2_ generations; however, in the F_3_ generation, the duration of first instar larval stage did not significantly differ between those feeding on rose flowers and 10% honey solution + kidney bean pods. In all three generations, the duration of the second instar larval stage feeding on rose petals was longer than that of larvae feeding on rose flowers and 10% honey solution + kidney bean pods, and there was no significant difference between the two food types. The prepupal stage of thrips showed no significant change among feeding on the three food types in the same generation. The duration of pupal development of WFT feeding on rose petals was the longest in the F_1_, F_2_, and F_3_ generation. The duration of pupal development of WFT feeding on rose flowers was not significantly different than that of WFT feeding on 10% honey solution + kidney bean pods in the F_1_ generation; whereas it was shorter than that of feeding on 10% honey solution + kidney bean pods in the F_2_ and F_3_ generations. The development of the immature stage in all generations showed the following trend based on the food type: rose flowers <10% honey solution + kidney bean pods < rose petals.

### 3.2. Effects of Different Foods on WFT Longevity and Reproduction in Successive Generations

Comparing generations, we could find that the longevity of WFT changed significantly after WFT transferred from kidney bean pods to the other three kinds of food (Table 2). When WFT fed on rose petals, the longevity of female adults was not significantly prolonged in the F_1_ generation, but it was significantly prolonged in F_2_ and F_3_ generations. The longevity of male adults did not significantly differ with the control diet in all three generations. Compared with the control, the longevities of female adults and male adults of WFT feeding on rose flowers and 10% honey solution + kidney bean pods were significantly prolonged since the F_1_ generation. When comparing the longevity of adults among the three kinds of food in the same generation, we could find similar results for the F_1_, F_2_, and F_3_ generations. The longevity of both female and male thrips feeding on rose flowers was significantly higher than that of thrips feeding on rose petals. The longevity of thrips feeding on 10% honey solution + kidney bean pods showed no significant difference compared with that of rose flowers, and there was also no significant difference with that of kidney bean pods.

The fecundity of female thrips feeding on rose petals was significantly increased in the F_1_ generation, and it continuously increased in the F_2_ generation. However, no significant difference was observed between the F_2_ and F_3_ generations. WFT had significantly higher oviposition of single females when WFT fed on rose flowers and 10% honey solution + kidney bean pods in the F_1_ generation compared to the control, and there were no differences among the F_1_, F_2_, and F_3_ generations.

In all three generations, fecundities of WFT showed a similar trend when thrips fed on the three types of food: rose flowers >10% honey solution + kidney bean pods > rose petals.

### 3.3. Effects of Different Foods on WFT Survival in Successive Generations

The age-stage specific survival curves (*s_xj_*) of WFT offspring feeding on different foods in F_1_, F_2_, and F_3_ generations were different (Figure 1). The *s_xj_* curve of thrips feeding on different foods overlapped less in each subsequent generation, indicating that the development rate of thrips in each growth stage was relatively consistent. At the immature stage, the survival rate of the second instar larvae was higher than that of first instar larvae, and that of the prepupal stage was lower than that of the pupal stage. The survival rate of the adult stage was <50%. The survival time of female adults was longer than that of male adults. Overall, the immature survival rates of WFT were higher when feeding on rose petals, rose flowers, and 10% honey solution + kidney bean pods compared to the control. Adult survival rates did not differ notably when feeding on different food types, and those of generations feeding on the same food did not differ significantly.

The age-specific survival rate (*l_x_*) curve of the population was the survival rate of the x age, age-specific fecundity (*m_x_*), age-stage-specific fecundity (*f_x_*), and reproduction value (*l_x_m_x_*) of all surviving individuals at age *x*, as shown in Figure 2. The early slope of the survival curve of WFT feeding on different foods was relatively flat, indicating that WFT mortality was lower in the immature stage. The survival time of WFT feeding on different foods in the same generation decreased in the following order: rose flowers >10% honey solution + bean pods > rose petals. The values of *m_x_*, *f_x_*, and *l_x_m_x_* under each treatment showed a trend of first increasing and then decreasing, with multiple peaks. Overall, the values were the highest for rose flowers and lowest for kidney bean pods. These values in thrips feeding on the same food did not change significantly among the different generations.

The age-stage life expectancy (*e_xj_*) represented the remaining survival time of individuals at age *x* and stage *j* (Figure 3). The WFT life expectancy decreased with age in different generations after feeding on different foods. A common feature was that the life expectancy of female adults was higher than that of male adults. In each generation, the life expectancy of thrips was highest when feeding on rose flowers, followed by 10% honey solution + kidney bean pods, and was the lowest when feeding on rose petals in the F_1_ generation. There was no notable difference between those feeding on rose petals and kidney bean pods.

The age-stage reproductive value (*v_xj_*) refers to the average contribution of individuals in the *x* age and *j* stage of WFT to the future population development (Figure 4). As shown in Figure 4, the reproductive peak of female adults of WFT appeared first when feeding on rose petals in the F_1_ generation; whereas, the reproductive peak of those feeding on 10% honey solution + kidney bean pods in the F_3_ generation appeared last, and there was little difference in the reproductive peak of female adults fed on the same food types among different generations. In addition, the *v_xj_* of WFT feeding on rose flowers was the highest in all three generations, followed by that of WFT feeding on 10% honey solution + kidney bean pods, and the lowest value was observed in WFT feeding on rose petals and kidney bean pods.

### 3.4. Effects of Different Foods on the Population Parameters of Successive Generations of WFT

The population parameters of the offspring of WFT feeding on different foods are shown in Table 3. The net increment rate (*R*_0_), intrinsic growth rate (*r*), and average generation cycle (*T*) in thrips fed on rose petals were not significantly different from those that fed on kidney bean pods in all three generations. However, *R*_0_, *r*, and finite rate of increase (*λ*) in thrips feeding on rose flowers and 10% honey solution + kidney bean pods were higher, and *T* was shorter than that of the control in all three generations. There was no significant difference among the F_1_, F_2_, and F_3_ generations when thrips fed on the same food.

Comparing the life parameters among the different food types in the same generation, it was found that *R*_0_, *r*, and *λ* of WFT fed on rose flowers and 10% honey solution + kidney bean pods were higher than those fed only on rose petals in the F_1_ and F_3_ generations. These values for thrips feeding on 10% honey solution + kidney bean pods were not significantly different from those for thrips feeding on rose petals in the F_2_ generation. The *T* value did not change significantly among the three food types in the F_1_ generation; whereas it was significantly shorter in thrips feeding on rose flowers than in those feeding on rose petals in the F_2_ generations. The *T* of thrips feeding on rose petals was significantly longer than that of thrips feeding on the other two food types in the F_3_ generation.

## 4. Discussion

It is generally believed that the short development time and high reproductive ability of insects indicate that insects feed on more nutritious food. In this study, we found that the development time of thrips (from immature to adult stages) feeding on different food types was different and increased in the following order: rose flowers <10% honey solution + kidney bean pods < rose petals < kidney bean pods. This may have been because flowers contain nectar and pollen, which provides more nutrients than other plant parts (unpublished data). This suggests that pollen and nectar have a more significant effect on the thrips than rose petals. Cao et al. [36] found that there was a significant positive correlation between the development rate of WFT and the soluble protein content of plant flowers. Gou et al. [37] found that *Bradysia cellarum* and *B. impatiens* displayed nutrient preference toward chives, and broad bean contained higher levels of protein, free amino acid, and soluble sugar and starch than other plants. Our findings were consistent with those of previous studies, suggesting that flowers play a key role in higher survival rates, shorter development, and higher fecundity, which leads to an increase in population at the flowering stage and affects the seasonal dynamics of the rose plant [26,38]. Moreover, after WFT fed on nutritious foods, they could more effectively digest and absorb nutrients [39]. Gerin et al. [19] found that the WFT population grew exponentially in the presence of flowers, which also demonstrated the effect of flower on stimulating population growth. Cotton pollen could reduce the developmental time of the immature stage of WFT [40]. Cao et al. [22] showed that WFT had a higher appetite for rose flower than for 17 other horticultural flowers. Cao et al. [23] found that the immature development time of WFT fed on roses was significantly shorter than that of WFT fed on gerbera. Zhu et al. [41] found that the nymph duration of *Cyrtorhinus lividipennis* was significantly reduced and the female adult stage was evidently prolonged after adults were fed on *Tagetes erecta*, *Trida procumbens*, *Emilia sonchifolia*, and *Sesamum indicum* flowers. Flower petals were used as the test materials to determine the effect on the development of thrips [23,24]; compared with these studies, we emphasize the role of single rose petal to a whole rose flower.

The results of this study showed that the immature development time of WFT significantly decreased in the F_1_ generation regardless of the food type, implying that the food types influenced thrips development and indicating that these three food types have higher nutritional value than that of kidney bean pods alone. The degrees of influence were different among the three food types. The development time of immature thrips feeding on rose flowers and 10% honey solution + kidney bean pods was shorter in the F_2_ generation, and there was no difference between the F_2_ and F_3_ generations, indicating that the immature development time of WFT fed on these two food types could reach a stable level after the F_2_ generation. However, the duration of immature development of thrips feeding on rose petals was shorter in the F_3_ generation compared with that in the F_1_ generation. These results showed that the changes in the development time were influenced by the food types and generation. Li et al. [42] found when *Apolygus lucorum* fed on corn and corn + cotton bollworm eggs, there was no significant difference in the total development duration of larvae among three consecutive generations, which contradicted the results of this study; this may have been because different insects have different adaptation mechanisms to food.

This study found that the adult longevity and fecundity of WFT feeding on the three food types were significantly higher than those fed on kidney bean pods alone, and that there was no significant difference among generations. Furthermore, WFT could adapt to a new food type after one generation of consuming it. It also found that in the same generation, female adults that fed on rose petals had the shortest life span, and there was no significant difference in the longevities of females and males that were fed rose flowers and 10% honey solution + kidney bean pods. In all generations, the fecundity with respect to food type decreased in the following order: rose flowers > 10% honey solution + kidney bean pods > rose petals, indicating that rose flowers were more conducive to increasing the WFT population. These results indicated that pollen and nectar contributed to increasing the WFT population and provide evidence of the positive effect of dietary honey on thrips. Previous studies have also shown a similar effect of pollen. When WFT were fed food containing pine pollen, the population was 22 times higher than that of thrips fed a diet without pine pollen [7]. Pollen addition as a dietary supplement had a positive effect on fecundity, presumably by improving the nutritional quality of the WFT diet [20,40,43]. Li et al. [44] also found that when *T. hawaiiensis* was fed food containing tea pollen, its fecundity was significantly improved and the adult lifespan was prolonged.

The life table parameters of *r* and *R*_0_ are important indicators and comprehensively reflect the growth, development, survival rate, reproduction, and population growth potential of insects under specific diets and environmental conditions [45,46,47]. This study found that compared to thrips feeding on kidney bean pods, the changes in *r* and *R*_0_ of thrips feeding on rose flowers and 10% honey solution + kidney bean pods were significantly higher in the F_1_ generation. In the same generation, the *r* and *R*_0_ of the population that fed on rose petals were generally lower than those of thrips that fed on the other two food types. This may have been because the rose flowers and 10% honey solution + kidney bean pods were more nutritious and were beneficial to the population performance, and feeding on these two food types led to an increase in several parameters. The differences in life table parameters in different foods were also likely determined by the nutritional value. The results also indicated that pollen, nectar, and honey (but not rose petal) had a significant effect on insect population parameters, which suggested that the different flower parts had different effects. In this regard, this study demonstrated that both rose flowers and 10% honey solution + kidney bean pods were more suitable for thrips reproduction than rose petals and kidney bean pods. This finding may explain why the highest thrips populations occur during the rose blooming period, when flowers containing pollen and nectar are available. Other studies have also demonstrated that flowers promote the fitness of insect pests [48] and that the appearance of flowers promoted thrips density [18]. Extrafloral nectar can provide an important contribution to population growth and maintenance of the predator *Iphiseius degenerans* on *Ricinus communis* [49]. Several previous studies also reported that pollen, nectar, and honey solution were added into food of insects and were more conducive to their growth and reproduction [44,50]. In our previous studies, we found that rose flowers and 10% honey solution + kidney bean pods had higher soluble sugar content and protein content, while these contents were lower in rose petals and kidney bean pods [51]. This indicted there were significant connections between life parameters and the different types of nutrients in the different parts of the plant.

The results provide evidence that honey also benefits thrips, and thrips supplied with honey solution exhibited higher population parameter values. In most cases, thrips feeding on 10% honey solution + kidney bean pods had a shorter developmental duration, longer female longevity, and a higher survival rate and fecundity than that of thrips feeding on kidney bean pods. The study confirmed earlier findings that supplemental foods improved the population performance. For example, van Rijn et al. [52] found that the adult longevity of *T. tabaci* feeding on honey solution was twice as long as that on cucumber leaf. Fu et al. [8] also found that supplying honey solution reduced the developmental time, increased adult longevity, and enhanced the fecundity of *T. hawaiiensis*.

## 5. Conclusions

In summary, our findings reveal that different rose flower parts have different effects on thrips, whereby pollen and nectar in flowers positively affect growth, development, and reproduction. This result explains why WFT breakouts occur after the rose flowering in the field. Therefore, controlling the thrips population prior to rose flowering will likely result in rose plants being more productive and have a positive effect on WFT management.

## Figures and Tables

**Figure 1 insects-14-00088-f001:**
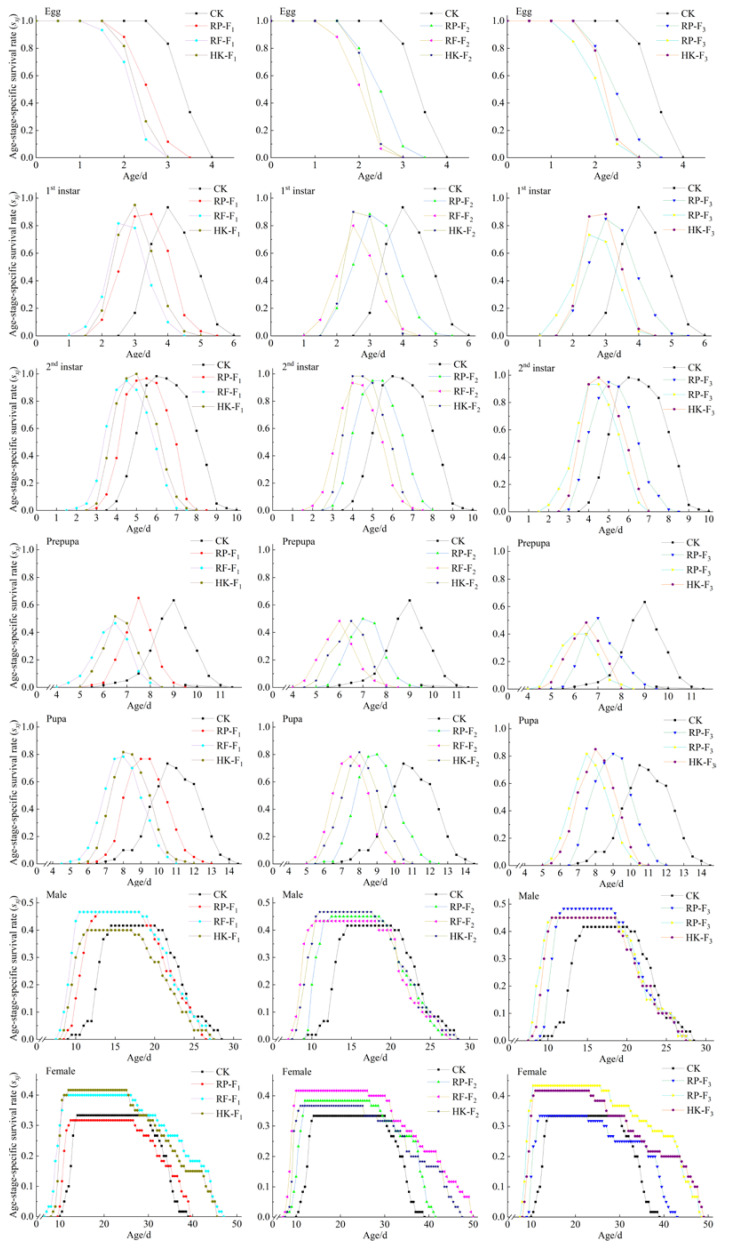
Age-stage specific survival rate (*s_xj_*) of *F. occidentalis* fed on rose petals (RP), rose flowers (RF), and 10% honey solution + kidney bean pods (HK) in F_1_, F_2_, and F_3_ generations, and control group *F. occidentalis* feeding on kidney bean pods (CK).

**Figure 2 insects-14-00088-f002:**
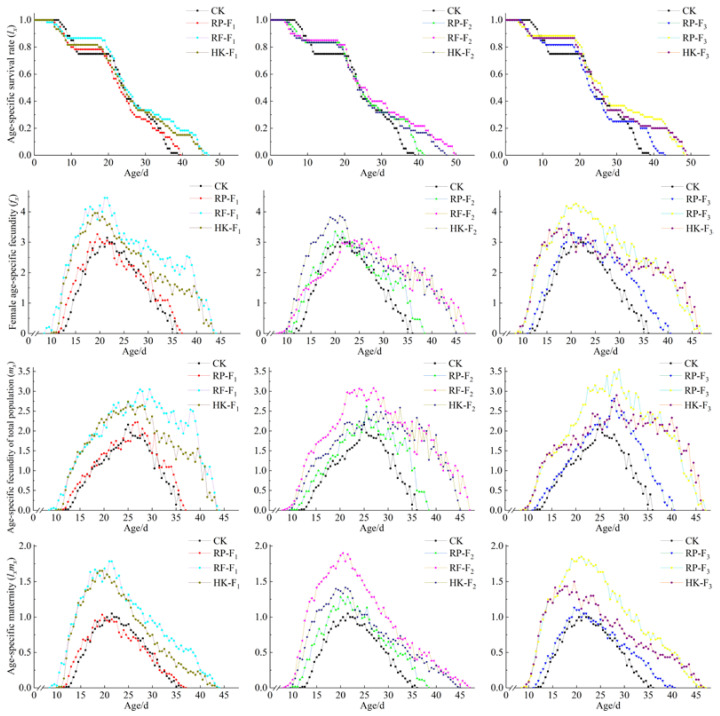
Age-specific survival rate (*l_x_*), female age-specific fecundity (*f_x_*), age-specific fecundity of total population (*m_x_*), and age-specific maternity (*l_x_m_x_*) of *F. occidentalis* fed on different food types in F_1_, F_2_, and F_3_ generations.

**Figure 3 insects-14-00088-f003:**
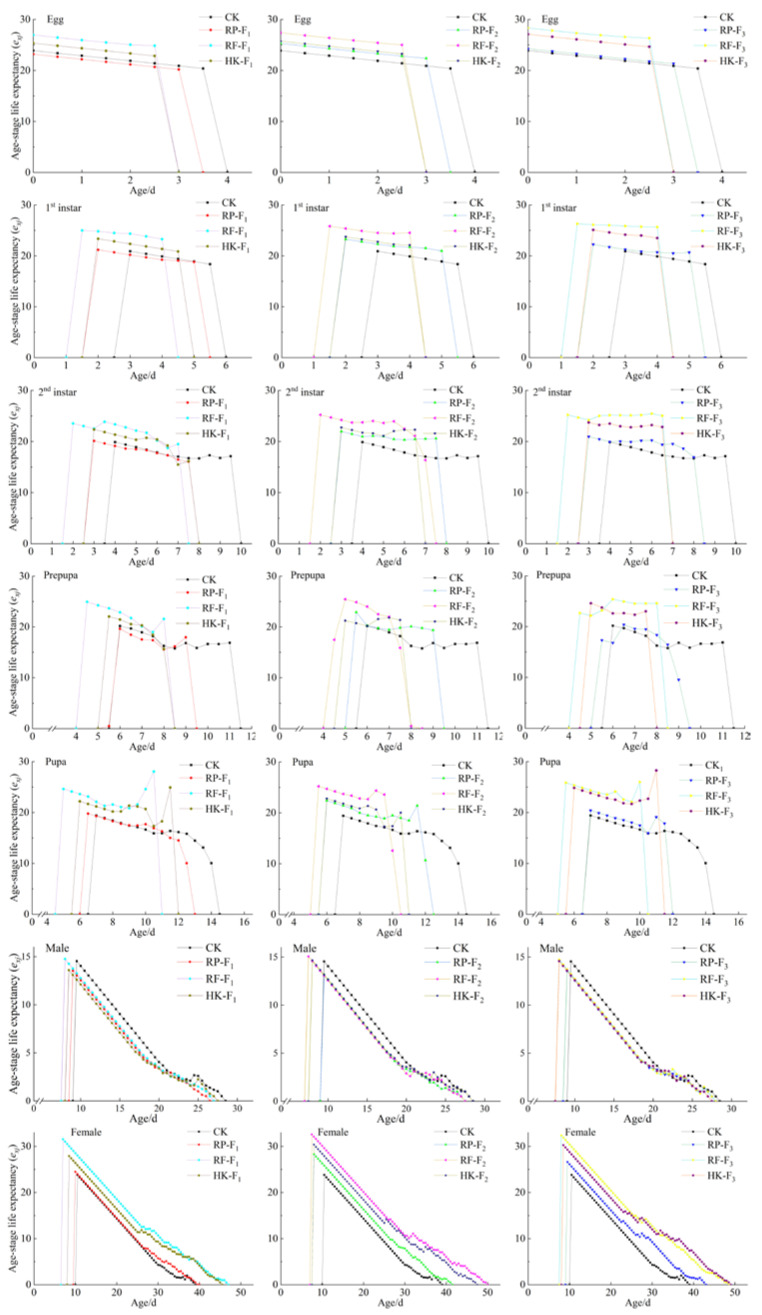
Age-stage life expectancy (*e_xj_*) of *F. occidentalis* on different food types in F_1_, F_2_, and F_3_ generations.

**Figure 4 insects-14-00088-f004:**
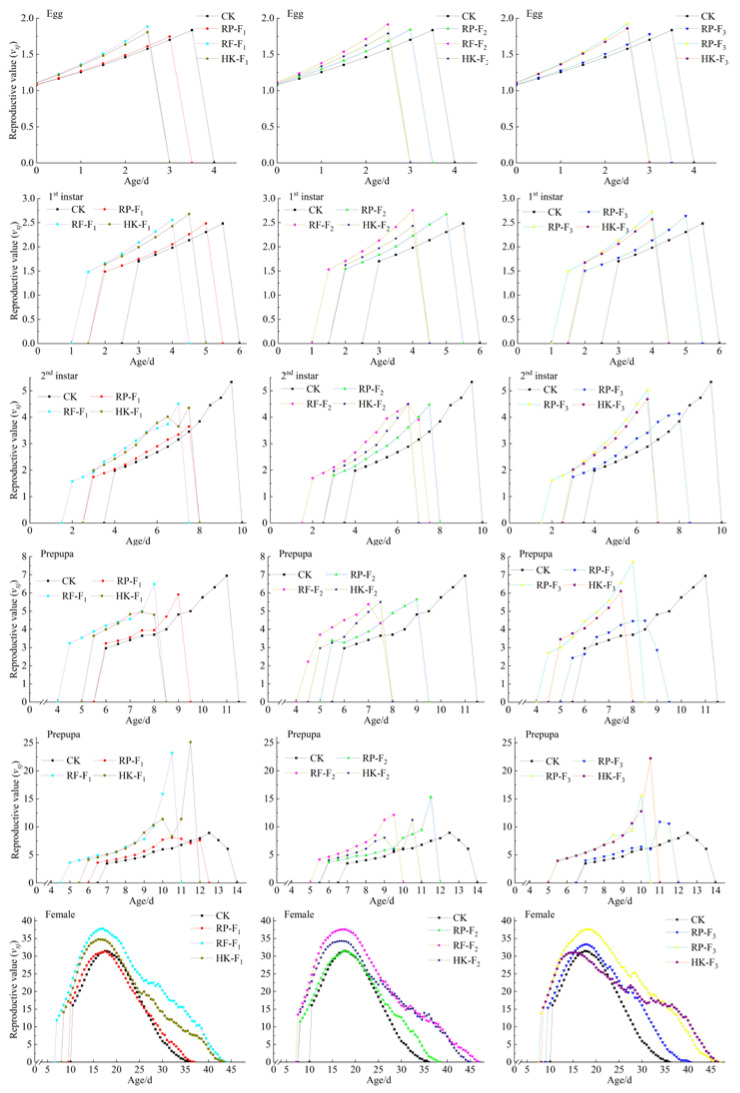
Age-stage reproductive value (*v_xj_*) of *F. occidentalis* on different food types in F_1_, F_2_, and F_3_ generations.

**Table 1 insects-14-00088-t001:** Effect of different foods on *F. occidentalis* development.

Stage	Generation	Rose Petals	Rose Flowers	10% Honey Solution + Kidney Bean Pods
Egg (d)	Control	3.58 ± 0.04 a	3.58 ± 0.04 a	3.58 ± 0.04 a
	F_1_	2.77 ± 0.05 bA	2.38 ± 0.05 bC	2.54 ± 0.04 bB
	F_2_	2.68 ± 0.06 bA	2.24 ± 0.05 bC	2.43 ± 0.04 bB
	F_3_	2.71 ± 0.06 bA	2.27 ± 0.06 bC	2.46 ± 0.04 bB
1st instar (d)	Control	1.52 ± 0.04 ab	1.52 ± 0.04 ab	1.52 ± 0.04 ab
	F_1_	1.57 ± 0.03 aA	1.21 ± 0.05 bC	1.37 ± 0.04 bB
	F_2_	1.49 ± 0.04 abA	1.12 ± 0.05 bC	1.23 ± 0.03 cB
	F_3_	1.46 ± 0.04 bA	1.15 ± 0.05 bB	1.25 ± 0.03 cB
2nd instar (d)	Control	3.30 ± 0.05 a	3.30 ± 0.05 a	3.30 ± 0.05 a
	F_1_	2.77 ± 0.06 bA	2.52 ± 0.05 bB	2.60 ± 0.05 bB
	F_2_	2.68 ± 0.06 bA	2.43 ± 0.04 bB	2.49 ± 0.04 bB
	F_3_	2.70 ± 0.06 bA	2.45 ± 0.05 bB	2.48 ± 0.05 bB
Prepupa (d)	Control	1.30 ± 0.05 a	1.30 ± 0.05 a	1.30 ± 0.05 a
	F_1_	1.03 ± 0.05 bA	0.91 ± 0.05 bA	0.93 ± 0.04 bA
	F_2_	0.98 ± 0.05 bA	0.87 ± 0.05 bA	0.89 ± 0.04 bA
	F_3_	1.02 ± 0.05 bA	0.90 ± 0.05 bA	0.90 ± 0.04 bA
Pupa (d)	Control	2.92 ± 0.06 a	2.92 ± 0.06 a	2.92 ± 0.06 a
	F_1_	2.72 ± 0.06 bA	2.39 ± 0.07 bB	2.51 ± 0.07 bB
	F_2_	2.65 ± 0.06 bA	2.24 ± 0.06 bC	2.40 ± 0.05 bB
	F_3_	2.68 ± 0.07 bA	2.27 ± 0.05 bC	2.46 ± 0.05 bB
Immature (d)	Control	12.64 ± 0.15 a	12.64 ± 0.15 a	12.64 ± 0.15 a
	F_1_	10.87 ± 0.12 bA	9.39 ± 0.12 bC	9.92 ± 0.11 bB
	F_2_	10.54 ± 0.12 bcA	8.86 ± 0.09 cC	9.46 ± 0.11 cB
	F_3_	10.54 ± 0.10 cA	9.09 ± 0.10b cC	9.55 ± 0.09 cB

“Immature” represents the time of thrips developing from egg to adult. Different lowercase letters after means (±SE) in the column in the same WFT stage indicate the significant differences among different generations of the same food, and different capital letters in the row indicate significant differences among different foods in the same generation (*p* < 0.05).

**Table 2 insects-14-00088-t002:** The effect of different food types on adult *F. occidentalis* longevity and reproduction.

Fecundity	Generation	Rose Petals	Rose Flowers	10% Honey Solution + Kidney Bean Pods
Longevity of Female (d)	Control	21.80 ± 0.44 b	21.80 ± 0.44 b	21.80 ± 0.44 b
	F_1_	23.47 ± 0.89 abB	29.08 ± 1.36 aA	26.37 ± 1.41 aAB
	F_2_	25.78 ± 0.85 aB	31.16 ± 1.38 aA	28.92 ± 1.47 aAB
	F_3_	25.55 ± 1.29 aB	31.22 ± 1.40 aA	29.12 ± 1.74 aAB
Longevity of Male (d)	Control	11.28 ± 0.16 a	11.28 ± 0.16 b	11.28 ± 0.16 b
	F_1_	11.75 ± 0.47 aB	13.43 ± 0.19 aA	12.33 ± 0.46 aAB
	F_2_	12.11 ± 0.44 aB	13.77 ± 0.51 aA	13.21 ± 0.50 aAB
	F_3_	12.04 ± 0.49 aB	13.71 ± 0.48 aA	13.11 ± 0.48 aAB
Eggs/female	Control	80.84 ± 1.40 c	80.84 ± 1.40 b	80.84 ± 1.40 b
	F_1_	86.89 ± 1.89 bC	155.69 ± 7.00 aA	124.82 ± 5.47 aB
	F_2_	99.08 ± 2.55 aC	163.17 ± 6.53 aA	135.80 ± 5.22 aB
	F_3_	101.24 ± 5.01 aC	168.27 ± 7.03 aA	138.37 ± 7.96 aB

Different lowercase letters after means (±SE) in the column in the same WFT stage indicate the significant differences among different generations of the same food, and different capital letters in the row indicate significant differences among different foods in the same generation (*p* < 0.05).

**Table 3 insects-14-00088-t003:** Life-table parameters of *F. occidentalis* offspring feeding on different foods.

Life Table Parameters	Generation	Rose Petals	Rose Flowers	10% Honey Solution + Kidney Bean Pods
Net reproductive rate (*R*_0_)	Control	26.95 ± 4.94 a	26.95 ± 4.94 b	26.95 ± 4.94 b
	F_1_	27.51 ± 5.25 aB	62.21 ± 10.23 aA	51.99 ± 8.28 aA
	F_2_	37.96 ± 6.29 aB	67.96 ± 10.73 aA	49.83 ± 8.67 aAB
	F_3_	33.75 ± 6.36 aB	72.97 ± 11.23 aA	57.64 ± 9.38 aA
Intrinsic rate of increase (*r*)	Control	0.1510 ± 0.0095 a	0.1510 ± 0.0095 b	0.1510 ± 0.0095 b
	F_1_	0.1581 ± 0.0105 aB	0.2052 ± 0.0100 aA	0.1964 ± 0.0090 aA
	F_2_	0.1724 ± 0.0092 aB	0.2138 ± 0.0097 aA	0.1929 ± 0.0103 aAB
	F_3_	0.1624 ± 0.0098 aB	0.2100 ± 0.0091 aA	0.2051 ± 0.0098 aA
Finite rate of increase (*λ*)	Control	1.1630 ± 0.0111 a	1.1630 ± 0.0111 b	1.1630 ± 0.0111 b
	F_1_	1.1714 ± 0.0123 aB	1.2278 ± 0.0123 aA	1.2171 ± 0.0110 aA
	F_2_	1.1882 ± 0.0110 aB	1.2385 ± 0.0120 aA	1.2129 ± 0.0125 aAB
	F_3_	1.1764 ± 0.0116 aB	1.2338 ± 0.0112 aA	1.2277 ± 0.0120 aA
Mean generation time (*T*)/d	Control	21.71 ± 0.33 a	21.71 ± 0.33 a	21.71 ± 0.33 a
	F_1_	20.85 ± 0.33 aA	20.08 ± 0.39 bA	20.05 ± 0.26 bA
	F_2_	21.02 ± 0.36 aA	19.68 ± 0.34 bB	20.19 ± 0.37 bAB
	F_3_	21.56 ± 0.37 aA	20.38 ± 0.36 bB	19.71 ± 0.36 bB

Different lowercase letters after means (±SE) in the column in the same WFT stage indicate the significant differences among different generations of the same food, and different capital letters in the row indicate significant differences among different foods in the same generation (*p* < 0.05).

## Data Availability

The data that support the findings of this study are available from the corresponding author upon reasonable request.

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
