# Peer review of "Effects of Different Parts of the Rose Flower on the Development, Fecundity, and Life Parameters of Frankliniella occidentalis (Pergande) (Thysanoptera: Thripidae)"

_insects, 2023, doi:10.3390/insects14010088_

Round 1

Reviewer 1 Report

Summary: This study measured the life history parameters of F. occidentalis over 4 generations when feeding on rose petals, rose flowers, honey, and kidney beans. This study aims to better understand why thrips may develop particularly well on rose flowers. In particular, the authors hypothesized that nectar and pollen play an important role in thrips development. 

Strengths: This study repeated this experiment over multiple generations and analyzed those data separately. 

Weaknesses: This study would be strengthened if the authors justify their use of these particular treatments as proxies for the role pollen and nectar in thrips development in rose flowers. How close are kidney beans to pollen in nutritional content (protein/lipid/sugar ratios? Particular types of amino acids?). It would have been a stronger paper if the authors used rose flowers, but removed the pollen and nectar and compared it with rose flowers containing pollen and nectar. I wonder how the other structures in the rose flower besides pollen and nectar (e.g. ovules) may affect thrips. While the authors argue that the changes in life history parameters is increased by the whole flower nutrient value, there also may be differences in humidity between these different treatment that are not addressed in this experimental design. In its current form, the experimental design fails to answer the questions asked. 

Specific comments 

Line 25 – s missing from rose petals and kidney bean pods 

Line 73-75 & Line 103: What does this single petal rose flower look like? Does it have all of the ovary still intact? This may also provide nutrient and microhabitat for thrips. 

Where there any differences among the kidney bean fed thrips generations during the F1, F2, and F3 time periods? Need to make sure they are consistent before combining. 

Line 195 – contraction on “wasn’t” needs to be written out as “was not” 

Figures – The lines/ colors are very hard to distinguish in these figures. Would it be possible distinguish the food type differently from the generations? Or just show differences between food type on one figure and generations in another? 

Line 320 – “emphasize role of each part of the flower” – This is an over reach on the experimental design that is comparing single petal to a “whole” flower- not each individual part of the flower. 

Author Response

Dear reviewer,

Re: Effects of Different Parts of the Rose Flower on the Develop-ment, Fecundity, and Life Parameters of Frankliniella occiden-talis (Pergande) (Thysanoptera: Thripidae)  (ID: insects-2122850)

We appreciate the instructive and positive comments for improvement of our manuscript from you. We thank you for instructions for us to approach the revision. we have carefully considered each comment from the reviewers (track changes mode). We have made revisions in response to them, as follows.

Yours sincerely,

All authors,

Institute of Entomology

Guizhou University

Reviewer Comments to Authors

  1. Line 25 – s missing from rose petals and kidney bean pods

Our response: Thanks for your suggestion. But considering whole rose petal or kidney bean pod as one kind of food, the singular form is used. For the uniformity in the text, the plural form will not be used, and we don’t add the s here.

  1. Line 73-75 & Line 103:

What does this single petal rose flower look like?

Our response: Sorry for that we didn’t describe it clearly. When the rose petal was used in the experiment for group rearing, the flowers were not open, and the flowers haven’t the nectar and pollen. The other parts of flower (including peduncle, receptacle, perianth, stamen, pistil) were taken away and just petals were kept (Line 73-75). When the rose petal was used in the individual experiment (Line 103), just one rose petal was used. When rose flower was used in the individual experiment (Line 103), which mean just one flower petal + all stamens and pistils in the flower as the test material.

  1. Does it have all of the ovary still intact? This may also provide nutrient and microhabitat for thrips.

Our response: Thank you for careful checks. What you say is right. For “rose flower”, it is a whole opening flower without any damage.

  1. Where there any differences among the kidney bean fed thrips generations during the F1, F2, and F3 time periods? Need to make sure they are consistent before combining.

Our response: We are very grateful for your opinions on this manuscript. The thrips reared with kidney bean pods for more than 50 generations were used for control, and their biology was stable. Then thrips were transferred to 3 kinds of foods, So, the differences among the thrips feeding kidney bean pods and F1, F2, and F3 generations feeding other 3 kinds of foods were compared.

  1. Line 195 – contraction on “wasn’t” needs to be written out as “was not”

Our response: Sorry for our carelessness, we have changed it.

  1. Figures – The lines/ colors are very hard to distinguish in these figures. Would it be possible distinguish the food type differently from the generations? Or just show differences between food type on one figure and generations in another?

Our response: Thanks for the suggestions. We have replotted the line charts and presented the results of each generation in a separate figure.

  1. Line 320 – “emphasize role of each part of the flower” – This is an over reach on the experimental design that is comparing single petal to a “whole” flower- not each individual part of the flower.

Our response: Thanks for the suggestion. We have revised the description.

Reviewer 2 Report

Insects MDPI

 Effects of different parts of the rose flower on the development, fecundity, and life parameters of Frankliniella occidentalis (Pergande) (Thysanoptera: Thripidae)

 Ding-Yin Li, Dan Zhou, Jun-Rui Zhi, Wen-Bo Yue and Shun-Xin Li

General comments:

In this study, the authors evaluate the influence of rose petals, rose flower and 10% honey solution 17 + bean pod on the growth, development and fecundity of F. occidentalis.

From my point of view, the article contributes little to generating new knowledge. The fact of comparing the rose petals vs. rose flowers vs. 10% honey solution + bean pod does not bring anything new. In addition to this, the writing throughout the document is very confusing. In many paragraphs I have suggested redrafting them and referencing the figures or tables in the text so that the writing is better understood.

Therefore, I have made a number of suggestions and consider that in the conditions that the writing is not ready to be published.

Detailed comments to the text

1. Introduction

In general terms, all the consulted bibliography (21-26) related to the influence of flowers on thrips development are already very old, except for some of them, such as number 8 and number 27.

2. Materials and Methods

L 90 = Rosa rugosa change to Rosa rugosa

L107= 2.2. The group-rearing of different foods of WFT

L116-119 = …Some of F1 adults were collected for the individual development and reproductive  experiment, the remaining F1 adults were inoculated into new box with three different  foods respectively, and allowed adults to lay eggs. After12 h oviposition of F1 adults, the  different foods with eggs were removed and placed in another clean insect box to continuously reared to F2 adults. It is not clear from this paragraph whether the F1 adults recovered for the next generation of F2 were fed the same type of food from which they previously emerged or were given another type of food. This must be clarified, since if it was the second case, this undoubtedly skewed the results obtained. The same observation for the next two generations (F2 and F3).

L124-125 = 2.3. Development duration and survival rate of WFT feeding on different foods for successive 124 generations

L132-137 = …After the eggs  hatched, the newly hatched first instar larvae were placed individually in a transparent  insect raising cup (bottom diameter, 2.9 cm; top diameter 3.7 cm; and height, 3.3 cm) with  four different foods and absorbent filter paper at the bottom of the cup. The development  time and survival of each instar were observed under the microscope and recorded twice  a day (8:00 a.m. and 20:00 p.m.), and fresh food was replaced every other day until thrips emerged into adults… Same comments as in the previous paragraph (L116-119).

L142 = 2.4. Adult longevity and fecundity of WFT in successive generations

 143-144 = …Newly emerged thrips were paired (one female and one male adult) and placed in a  transparent insect cup with four different foods… The authors do not mention the origin of these thrips.,

L147-148 = …The number of larvae hatched represented the number of eggs laid (Wait et al., 1934). Justify why the authors did not consider the number of eggs laid (fertility).

L148-149 =…The hatched  larvae were reared by different foods into the adult to sex determination. Same comment as in the paragraph on lines 116-119.

3. Results

L168-169 = 3.1. Effects of different foods on the growth, development, and reproduction of WFT in successive generations. I ask the authors that the subtitles of the results coincide with the subtitles of materials and methods, in this case they do not coincide and it is difficult to be looking for which section of materials and methods the authors refer to in this section of results.

L170-178 = When WFT fed on the same food,… decrease was not different among different foods. The writing of this paragraph is very confusing, it is not possible to relate the text with the data in table 1. I suggest redrafting this paragraph or reorganizing the table so that it is consistent with the text and is easily digestible when reading it.

L179 = Table 1. The effect of different foods on the development of F. occidentalis. The data in the table is not known what it is? Days, minutes etc. They also do not indicate what CK stands for. And by Immature what the authors mean.

L183- 200 = As for WFT feeding different foods … pod < rose petal. The same happens with this paragraph, the authors do not rely on any table or figure and thus it is difficult to understand their results. I also suggest redrafting this paragraph and relying on a figure or data table.

3.2. Effects of different foods on t the longevity and reproduction of WFT in successive generations

L202-204 = … When WFT fed on rose petal, the longevity of female adults was significantly prolonged after the F2 generation, while the longevity of male adults did not significantly  differ with control diet among the three generations (Table 2)… When reading this text, there is no relationship with what is seen in table 2. At least I do not see in the table what the authors say in the text and this is probably because the table is very confusing.

L204-205 = …Compared with the control, the longevities… in the F1 generation. Where is the control in the table? I suggest to the authors to put the control in the table, otherwise it is not possible to visualize what the authors mention in the text.

L207-2011 = … When compared the longevity of adult among three kinds of food in the same generation, 207 we could find that the longevity of both female and male thrips feeding on rose flower 208 were higher than that of thrips feeding on rose petal. The longevity of thrips feeding on 209 10% honey solution + kidney bean pod showed no significant difference compared with the other two foods. The wording is very confusing and I cannot see what the authors say in the text with what is in the box. It is necessary to redraft the text and make the chart more friendly to be able to understand it.

3.3. Effects of different foods on the survival and reproduction rates of WFT in successive 221 generations

L223-233 = The age-stage specific survival curve… same food did not differ significantly.  I suggest to the authors that when describing the results in the text they indicate the figures corresponding to each biological state of the thrips, otherwise it is difficult to see the results..

L235 and 237 = Italicize Frankliniella occidentalis

L238 -248 = The age-specific survival rate (lx) curve … the different generations. Same suggestion as in the text of lines 223-233.

 L251 = Italicize F. occidentalis

L253-260 = The age-stage life expectancy (exj)… petal and kidney bean pod. Same suggestion as in the text of lines 223-233.

L262 = Italicize F. occidentalis

L264-272 = The age-stage reproductive value (vxj) refers… on rose petal and kidney bean pod. Same suggestion as in the text of lines 223-233.

L274 = Italicize F. occidentalis

3.4. Effects of different foods on the population parameters of successive generations of WFT

L285 = Italicize F. occidentalis

L 293 = …”the” change for “The”

4. Discussion

L358 = feedingrose cambar a feeding rose

L374 = Italicize Ricinus communis

The discussion contained in this work does not provide significant new knowledge, since it is already well known that nectar and pollen contribute to the quality of life of insects by favorably influencing their life parameters such as growth, development, survival rate, reproduction. and growth potential of the insect population. The fact of comparing the rose petals vs. rose flowers vs. 10% honey solution + bean pod does not really contribute anything new, perhaps if the authors made a proximal analysis of the three types of food evaluated and contrasted the sugar, protein, etc. contents, they could give their work a plus, but as it is, it does not contribute much to knowledge..

When the authors say that “The differences in life table parameters in different foods were also likely determined by the nutrition”. It is very obvious, but here it would be worth doing a proximal analysis to have the elements of what are the differences (if any) between the different types of nutrients in the different parts of the plant.

Author Response

Dear reviewer,

Re: Effects of Different Parts of the Rose Flower on the Develop-ment, Fecundity, and Life Parameters of Frankliniella occiden-talis (Pergande) (Thysanoptera: Thripidae)  (ID: insects-2122850)

We appreciate the instructive and positive comments for improvement of our manuscript from you. We thank you for instructions for us to approach the revision. we have carefully considered each comment from you (track changes mode). We have made revisions in response to them, as follows.

We hope the changes to meet your requirements for publication.

Thank you for your help.

Yours sincerely,

All authors,

Institute of Entomology

Guizhou University

Detailed comments to the text

  1. Introduction

In general terms, all the consulted bibliography (21-26) related to the influence of flowers on thrips development are already very old, except for some of them, such as number 8 and number 27.

Our response: Thanks for your suggestion. We updated some literatures in the introduction section, replacing the older literature by the latest literature. However, literature on the effects of pollen and nectar on the development of Frankliniella occidentalis is scarce in recent years and we have to refer to older literature.

  1. Materials and Methods

L 90 = Rosa rugosa change to Rosa rugosa

Our response: Sorry for our carelessness, we have changed it.

L107= 2.2. The group-rearing of different foods of WFT

L116-119 = Some of F1 adults were collected for the individual development and reproductive experiment, the remaining F1 adults were inoculated into new box with three different foods respectively, and allowed adults to lay eggs. After12 h oviposition of F1 adults, the different foods with eggs were removed and placed in another clean insect box to continuously reared to F2 adults. It is not clear from this paragraph whether the F1 adults recovered for the next generation of F2 were fed the same type of food from which they previously emerged or were given another type of food. This must be clarified, since if it was the second case, this undoubtedly skewed the results obtained. The same observation for the next two generations (F2 and F3).

Our response: Thanks for the suggestion. We have revised the description.

L124-125 = 2.3. Development duration and survival rate of WFT feeding on different foods for successive generations

L132-137 = After the eggs hatched, the newly hatched first instar larvae were placed individually in a transparent insect raising cup (bottom diameter, 2.9 cm; top diameter 3.7 cm; and height, 3.3 cm) with four different foods and absorbent filter paper at the bottom of the cup. The development time and survival of each instar were observed under the microscope and recorded twice a day (8:00 a.m. and 20:00 p.m.), and fresh food was replaced every other day until thrips emerged into adults… Same comments as in the previous paragraph (L116-119).

Our response: Thanks for the suggestion. We have revised the description.

L142 = 2.4. Adult longevity and fecundity of WFT in successive generations

L143-144 = Newly emerged thrips were paired (one female and one male adult) and placed in a transparent insect cup with four different foods… The authors do not mention the origin of these thrips.,

Our response: Sorry for that we didn’t describe it clearly. The origin of these thrips are from the adult thrips of section 2.3 (including F1, F2, F3 generations after feeding on three foods and control). We have added it.

L147-148 = The number of larvae hatched represented the number of eggs laid (Wait et al., 1934). Justify why the authors did not consider the number of eggs laid (fertility).

Our response: Thanks for the suggestion. And sorry for that we wrote author name wrong and recited the reference by wrong form. Because the thrips deposit the eggs in the tissue of the plant, the eggs cannot be observed directly. And Wait et al (1934) found that number of larvae hatched could represent the number of eggs laid, which was the fertility. This method adopted by most researchers on thrips. Another way of determining the number of eggs is by dyeing. It is too much work for us to finish. So, we used the method of Wait et al (1934).

L148-149 = The hatched larvae were reared by different foods into the adult to sex determination. Same comment as in the paragraph on lines 116-119.

Our response: Thanks for the suggestion. We have revised the description.

  1. Results

L168-169 = 3.1. Effects of different foods on the growth, development, and reproduction of WFT in successive generations. I ask the authors that the subtitles of the results coincide with the subtitles of materials and methods, in this case they do not coincide and it is difficult to be looking for which section of materials and methods the authors refer to in this section of results.

Our response: We agree with you suggestion and have modified the description in materials and methods section.

L170-178 = When WFT fed on the same food, … decrease was not different among different foods. The writing of this paragraph is very confusing, it is not possible to relate the text with the data in table 1. I suggest redrafting this paragraph or reorganizing the table so that it is consistent with the text and is easily digestible when reading it.

Our response: Thanks for the suggestion. We have revised the description.

L179 = Table 1. The effect of different foods on the development of F. occidentalis. The data in the table is not known what it is? Days, minutes etc. They also do not indicate what CK stands for. And by Immature what the authors mean.

Our response: Thanks for your careful checks. We are sorry for our carelessness. We calculate the development time of each stage in thrips in days and we have added to the table. “Immature” represents the time of thrips developing from egg to adult.

L183- 200 = As for WFT feeding different foods … pod < rose petal. The same happens with this paragraph, the authors do not rely on any table or figure and thus it is difficult to understand their results. I also suggest redrafting this paragraph and relying on a figure or data table.

Our response: Sorry for missing the key point and causing the difficult to understand. The results were also presented in the table 1. The differences among different foods were labeled by different capital letters in the same row.

3.2. Effects of different foods on the longevity and reproduction of WFT in successive generations

L202-204 = … When WFT fed on rose petal, the longevity of female adults was significantly prolonged after the F2 generation, while the longevity of male adults did not significantly differ with control diet among the three generations (Table 2) … When reading this text, there is no relationship with what is seen in table 2. At least I do not see in the table what the authors say in the text and this is probably because the table is very confusing.

Our response: In the table 2, different lowercase letters in the column in the same WFT stage indicate the significant differences among different generations at same food, which was the same with table 1. When WFT fed on kidney bean pod and rose petal in F1, F2 and F3 generation, the means (± SE) were 21.80±0.44b, 23.47±0.89ab, 25.78±0.85a, and 25.55±1.29a respectively. The control and F1 generation were labeled the same letter b, it was thought the difference wasn’t significant. But there were the different letters labeled after the control (b) and F2 (a) and F3(a) generation, so we said that the longevity of female adults significantly prolonged after the F2 generation. While the longevities of male adults in the in control and F1, F2 and F3 generation were labeled the same letter of a, so we said that the longevity of male adults did not significantly differ among control diet and the three generations.

L204-205 = …Compared with the control, the longevities… in the F1 generation. Where is the control in the table? I suggest to the authors to put the control in the table, otherwise it is not possible to visualize what the authors mention in the text.

Our response: Sorry for that we didn’t describe it clearly. We have replaced the CK by control to make it clearer.

L207-2011 = … When compared the longevity of adult among three kinds of food in the same generation, 207 we could find that the longevity of both female and male thrips feeding on rose flower 208 were higher than that of thrips feeding on rose petal. The longevity of thrips feeding on 209 10% honey solution + kidney bean pod showed no significant difference compared with the other two foods. The wording is very confusing and I cannot see what the authors say in the text with what is in the box. It is necessary to redraft the text and make the chart more friendly to be able to understand it.

Our response: In the table 2, different capital letters in the row indicate the significant differences among different foods in the same generation, which was the same with table 1. In both female and male longevities, the A was labeled after rose flower, and B was labeled after rose petal for all three generations, which indicated the significant difference. According to the value, we described that longevity of both female and male thrips feeding on rose flower were higher than that of thrips feeding on rose petal. The AB was labeled after 10% honey solution + kidney bean pod, which was not different with the after rose flower (labeling A ) and after rose petal (labeling B). So we described that the longevity of thrips feeding on 10% honey solution + kidney bean pod showed no significant difference compared with the other two foods.

3.3. Effects of different foods on the survival and reproduction rates of WFT in successive 221 generations

L223-233 = The age-stage specific survival curve… same food did not differ significantly. I suggest to the authors that when describing the results in the text they indicate the figures corresponding to each biological state of the thrips, otherwise it is difficult to see the results.

Our response: We have split all the graphs into three subgraphs to more clearly present the results.

L235 and 237 = Italicize Frankliniella occidentalis

Our response: Sorry for our carelessness, we have corrected it.

L238 -248 = The age-specific survival rate (lx) curve … the different generations. Same suggestion as in the text of lines 223-233.

Our response: We have split all the original line graphs into three subgraphs to more clearly present the results.

L251 = Italicize F. occidentalis

Our response: Sorry for our carelessness, we have corrected it.

L253-260 = The age-stage life expectancy (exj)… petal and kidney bean pod. Same suggestion as in the text of lines 223-233.

Our response: We have split all the graphs into three subgraphs to more clearly present the results.

L262 = Italicize F. occidentalis

Our response: Sorry for our carelessness, we have corrected it.

L264-272 = The age-stage reproductive value (vxj) refers… on rose petal and kidney bean pod. Same suggestion as in the text of lines 223-233.

Our response: We have split all the graphs into three subgraphs to more clearly present the results.

L274 = Italicize F. occidentalis

Our response: Sorry for our carelessness, we have corrected it.

3.4. Effects of different foods on the population parameters of successive generations of WFT

L285 = Italicize F. occidentalis

L 293 = …”the” change for “The”

Our response: Sorry for our carelessness, we have corrected it.

  1. Discussion

L358 = feedingrose cambar a feeding rose

L374 = Italicize Ricinus communis

Our response: Sorry for our carelessness, we have changed them.

The discussion contained in this work does not provide significant new knowledge, since it is already well known that nectar and pollen contribute to the quality of life of insects by favorably influencing their life parameters such as growth, development, survival rate, reproduction. and growth potential of the insect population. The fact of comparing the rose petals vs. rose flowers vs. 10% honey solution + bean pod does not really contribute anything new, perhaps if the authors made a proximal analysis of the three types of food evaluated and contrasted the sugar, protein, etc. contents, they could give their work a plus, but as it is, it does not contribute much to knowledge.

When the authors say that “The differences in life table parameters in different foods were also likely determined by the nutrition”. It is very obvious, but here it would be worth doing a proximal analysis to have the elements of what are the differences (if any) between the different types of nutrients in the different parts of the plant.

Our response: We are very grateful for your opinions on this manuscript. We have applied the information of the nutrition of different parts of the plant as you suggested.  We have rewritten this section

Round 2

Reviewer 1 Report

Line 13-14 – ‘s’ at the end of treatment types. While treatment names may be ‘rose petal’, ’rose flower’ and ‘kidney bean pod’, when you discuss the implications of your work, you are talking about the aggregate of all rose petals, rose flowers and kidney bean pods in general on WFT, thus using the plural is more correct

Line 76-77 – Still unclear why honey was included as a ‘supplemental food’. Justifying this treatment would greatly improve the manuscript. What sugars are present in honey? How does this compare to rose nectar? What is known about the effect of sugar on thrips life history? 

Line 83 – move the scientific name of kidney beans to the first mention in the sentence. 

Line 90 – “used”

Line 101 - “in the thrips individual development experiment”

Line 111, 118, 131 – missing a space before 12h

Line 127 – “adults reared on kidney bean pod”

In figures 1 & 2 – is there a way to include a measure of variance around the survival rates while still making the figures readable? 

In all figures -  I suggest making the treatments all the same colors for the different generations now that the generations have been split into different panels, which would decrease the number of legends necessary, and labeling the panels. Perhaps increasing the size of the points and lines would also make these figures even more legible.

Lines 382-383 – There is a large body of literature on supplemental pollen contributing to increased predator effectiveness and lowered herbivore populations. It is worth mentioning that the pollen and nectar of flowers may complicate predator-prey dynamics for WFT.  

Author Response

Dear reviewer,

Re: Effects of Different Parts of the Rose Flower on the Develop-ment, Fecundity, and Life Parameters of Frankliniella occidentalis (Pergande) (Thysanoptera: Thripidae) (ID: insects-2122850)

We highly appreciate the instructive and positive comments for improvement of our manuscript from you. We thank you for instructions for us to approach the revision. We have carefully considered each comment. We have made revisions in response to them, as follows.

Yours sincerely,

All authors,

Institute of Entomology

Guizhou University

  1. Line 13-14 – ‘s’ at the end of treatment types. While treatment names may be ‘rose petal’, ’rose flower’ and ‘kidney bean pod’, when you discuss the implications of your work, you are talking about the aggregate of all rose petals, rose flowers and kidney bean pods in general on WFT, thus using the plural is more correct.

Our response: Thanks for the suggestion, we have changes them in plural form in the text.

  1. Line 76-77 – Still unclear why honey was included as a ‘supplemental food’. Justifying this treatment would greatly improve the manuscript. What sugars are present in honey? How does this compare to rose nectar? What is known about the effect of sugar on thrips life history?

Our response: We are very grateful for your opinions on this manuscript. There is higher soluble sugar content in the honey, and the nectar also has the higher soluble sugar (Reference 51, Zhou, 2021), so we use the honey as supplemental food. Compared with the control (kidney bean pods), 10% honey solution + kidney bean pods had a significate influence on the development, fecundity and populate parameters, from which we know about the effect of sugar on thrips life history.

  1. Line 83 – move the scientific name of kidney beans to the first mention in the sentence.

Our response: Thanks for the suggestion, we have changed it.

  1. Line 90 – “used”

Our response: Sorry for our carelessness, we have changed it.

  1. Line 101 - “in the thrips individual development experiment”

Our response: Thanks for the suggestion, we have added it.

  1. Line 111, 118, 131 – missing a space before 12h

Our response: Sorry for our carelessness, we have corrected it.

  1. Line 127 – “adults reared on kidney bean pod”

Our response: Sorry for our carelessness, we have corrected it.

  1. In figures 1 & 2 – is there a way to include a measure of variance around the survival rates while still making the figures readable?

Our response: Thanks for the suggestion. However, it is impossible to do that.

  1. In all figures - I suggest making the treatments all the same colors for the different generations now that the generations have been split into different panels, which would decrease the number of legends necessary, and labeling the panels. Perhaps increasing the size of the points and lines would also make these figures even more legible.

Our response: We initially showed the different generations in one panel, but it was so dense and it was unreadable. Moreover, we tried to increase the size of the points and lines, but which still resulted in polylines stacked and unreadable. So, we don’t change the figure.

  1. Lines 382-383 – There is a large body of literature on supplemental pollen contributing to increased predator effectiveness and lowered herbivore populations. It is worth mentioning that the pollen and nectar of flowers may complicate predator-prey dynamics for WFT.

Our response: Thanks for the suggestion. We have revised the description.

Reviewer 2 Report

Dear Ms. Alyssa Kang

 Section Managing Editor, MDPI

Insects MDPI

 Effects of different parts of the rose flower on the development, fecundity, and life parameters of Frankliniella occidentalis (Pergande) (Thysanoptera: Thripidae)

 Ding-Yin Li, Dan Zhou, Jun-Rui Zhi, Wen-Bo Yue and Shun-Xin Li

Regarding the revised version of the manuscript "Effects of Different Parts of the Rose Flower on the Development, Fecundity, and Life Parameters of Frankliniella occidentalis (Pergande) (Thysanoptera: Thripidae)", which the authors resubmitted, I would like to resubmit my comments. and suggestions

Reference 24 is not referenced in the text: Cao, Y.; Yang, H.; Li, J.; Zhang, G.Z.; Wang, Y.W.; Li, C.; Gao, Y.L. Population development of Frankliniella occidentalis and Thrips hawaiiensis in constant and fluctuating temperatures. J. Appl. Entomol. 2019; 143, 49–57.

Neither does reference 27: Shen, D.R.; Zhang, H.R.; Li, Z.Y.; He, S.Y. Effects of different foods on the growth and development of Frankliniella occidentalis. Plant Prot. 2012, 38, 55-59.

Perhaps when changing the numbers of the references there was a mismatch in the numbering, this must be reviewed and corrected for all the references.

L149-152 = I already mentioned it in the previous review: fecundity and fertility must be obtained in this type of work. Here it is not clear if they obtained both parameters.

L168-169 = 3.1. Effects of different foods on the growth, development, and reproduction of WFT in successive generations. The authors ignored this suggestion.

 L 210 = add point “ generations Compared “!

L207-2011 = The authors did not make the modifications that were requested.

L223-233 = They did not make these modifications either.

L384 = Changer precious to previous.

L527-530 = Separate references 51 and 52.

L527 = Italicize Frankliniella occidentalis

Author Response

Dear reviewer,

Re: Effects of Different Parts of the Rose Flower on the Develop-ment, Fecundity, and Life Parameters of Frankliniella occiden-talis (Pergande) (Thysanoptera: Thripidae)  (ID: insects-2122850)

Thank you again for your instructive and positive comments for improvement of our manuscript from you. We have carefully considered each comment from you. We have made revisions in response to them, as follows.

Yours sincerely,

All authors,

Institute of Entomology

Guizhou University

  1. Reference 24 is not referenced in the text: Cao, Y.; Yang, H.; Li, J.; Zhang, G.Z.; Wang, Y.W.; Li, C.; Gao, Y.L. Population development of Frankliniella occidentalis and Thrips hawaiiensis in constant and fluctuating temperatures. J. Appl. Entomol. 2019; 143, 49–57.

Our response: Thanks for your careful checks. We are sorry for our carelessness, we have corrected it.

  1. Neither does reference 27: Shen, D.R.; Zhang, H.R.; Li, Z.Y.; He, S.Y. Effects of different foods on the growth and development of Frankliniella occidentalis. Plant Prot. 2012, 38, 55-59.

Our response: Sorry for our carelessness, we have corrected it.

  1. Perhaps when changing the numbers of the references there was a mismatch in the numbering, this must be reviewed and corrected for all the references.

Our response: Thanks for your careful checks. We are sorry for our carelessness, we have checked the references to match the numbers cited in the text.

  1. L149-152 = I already mentioned it in the previous review: fecundity and fertility must be obtained in this type of work. Here it is not clear if they obtained both parameters.

Our response: We are very grateful for your opinions on this manuscript. Maybe we didn’t describe it clearly. Because the thrips deposit the eggs in the tissue of the plant, we used the method of Watts et al (1934) to estimate thrips fecundity, in which he demonstrated that the number of larvae hatched represented the number of eggs laid. This method was used in the similar researches, such as the references cited (Cao et al, 2018; 2019) in the MS. In the MS, fecundity is represented by ‘Eggs/female’ in Table 2. It is difficulty to exactly determine the percentage of the eggs hatch, so the fertility can’t be gotten because of thrips special deposition mode.

  1. L168-169 = 3.1. Effects of different foods on the growth, development, and reproduction of WFT in successive generations. The authors ignored this suggestion.

Our response: We agree with you suggestion, and sorry for our carelessness. We have changed 3.1 subtitle.

  1. L 210 = add point “generations Compared”!

Our response: Sorry for our carelessness, we have corrected it.

  1. L207-2011 = The authors did not make the modifications that were requested.

Our response: We have revised again the description based on your suggestion.

L223-233 = They did not make these modifications either.

Our response: We have revised again the description about L223-233. As for the age-stage specific survival curve, there were two aspects comparison. We compared the differences among different generations at same food, and we also compared the differences among different foods in the same generation. If we described the results in the text, they indicate the figures corresponding to each biological state of the thrips, the number of figures will be doubled. For clearer, we have split the figures in last version. but it is difficulty to double figures due to limit in space.

L384 = Changer precious to previous.

Our response: Thanks again. We have changed it.

  1. L527-530 = Separate references 51 and 52.

Our response: Sorry for our carelessness, we have separated them.

  1. L527 = Italicize Frankliniella occidentalis

Our response: Sorry for our carelessness, we have corrected it.